# SARS-CoV-2 Spike Does Not Possess Intrinsic Superantigen-like Inflammatory Activity

**DOI:** 10.3390/cells11162526

**Published:** 2022-08-15

**Authors:** Carola Amormino, Valentina Tedeschi, Giorgia Paldino, Stefano Arcieri, Maria Teresa Fiorillo, Alessandro Paiardini, Loretta Tuosto, Martina Kunkl

**Affiliations:** 1Department of Biology and Biotechnology Charles Darwin, Sapienza University, 00185 Rome, Italy; 2Laboratory Affiliated to Istituto Pasteur Italia-Fondazione Cenci Bolognetti, Sapienza University, 00185 Rome, Italy; 3Department of Surgical Sciences, Sapienza University of Rome, 00185 Rome, Italy; 4Department of Biochemical Sciences “A. Rossi Fanelli”, Sapienza University of Rome, 00185 Rome, Italy

**Keywords:** SARS-CoV-2 spike, SEB, CD4^+^ T cells, superantigen, MIS-C

## Abstract

Multisystem inflammatory syndrome in children (MIS-C) is a rare hyperinflammatory disease occurring several weeks after SARS-CoV-2 infection. The clinical similarities between MIS-C and the toxic shock syndrome, together with the preferential expansion of T cells with a T-cell receptor variable β chain (TCRVβ) skewing, suggested a superantigen theory of MIS-C. For instance, recent in silico modelling evidenced the presence of a highly conserved motif within SARS-CoV-2 spike protein similar in structure to the superantigenic fragment of staphylococcal enterotoxin B (SEB). However, experimental data on the superantigenic activity of the SARS-CoV-2 spike have not yet been provided. Here, we assessed the superantigenic activity of the SARS-CoV-2 spike by analysing inflammatory cytokine production in both Jurkat cells and the peripheral blood CD4^+^ T cells stimulated with the SARS-CoV-2 spike or SEB as a control. We found that, unlike SEB, the SARS-CoV-2 spike does not exhibit an intrinsic superantigen-like activity.

## 1. Introduction

Severe acute respiratory syndrome coronavirus 2 (SARS-CoV-2) is a beta-coronavirus that causes the respiratory syndrome known as COVID-19 [1,2]. Although most infected people develop a mild disease characterised by fever, dry cough, and gastrointestinal (GI) symptoms, around 20% of patients progress to severe pneumonia with extensive lung damage and breathing difficulties, which in some cases lead to respiratory failure and death [3]. Multisystem inflammatory syndrome (MIS), caused by the hyperactivation of the immune system and the release of excessive inflammatory cytokines, a process known as cytokine storm, has also been observed in some COVID-19 patients resulting in death in some of them [4,5,6]. Notably, MIS also occurs in a small proportion of SARS-CoV-2-infected children (MIS-C), especially in male children, who, within a few weeks from infection, develop severe hyperinflammatory disease with multiorgan dysfunctions [7,8,9].

Children affected by MIS-C manifest a persistent fever, cardiovascular dysfunctions, and severe GI, respiratory, and neurological symptoms [10,11,12,13,14,15,16,17]. The main immunological features of MIS-C involve the production of high numbers of inflammatory cytokines and chemokines, a strong T-cell activation with the preferential expansion of T cells with T-cell receptor variable β chain (TCRVβ) skewing [18,19,20,21,22,23]. The strong similarity between MIS-C and toxic shock syndrome (TSS) elicited by bacterial superantigens (SAgs) [24], together with the recent computational identification of a sequence within the spike of SARS-CoV-2 that is similar in structure to *Staphylococcus aureus* enterotoxin B (SEB) [19,25], prompted us to postulate an SAg activity of the SARS-CoV-2 spike [26,27].

SEB is a staphylococcal SAg that, by binding to the outer leaflet of major histocompatibility class II (MHC-II) molecules on antigen-presenting cells (APCs) and specific TCRVβ chain elements, induces the polyclonal activation of T cells [28,29,30,31,32,33] with high levels of inflammatory cytokines, including TNF-α, IL-2 and IFN-γ, which contribute to respiratory failure, multiorgan system breakdown, and death [32,33]. To induce optimal inflammatory cytokine production, in addition to TCR, SEB also binds to CD28 [34,35,36,37,38], an important costimulatory molecule [39,40] that, in human T cells, triggers TCR-independent inflammatory signals [41,42,43,44,45].

Recent in silico studies evidenced that the SARS-CoV-2 spike contains a sequence (T_678_ to Q_690_) enclosing a unique polybasic insert (P_681_RRARS_686_), which is highly conserved in all of the SARS-CoV-2 variants [46] and that shares similar features with the superantigenic region of SEB (T_130_ to D_161_) [19]. Interestingly, the same superantigenic fragment of SEB is also involved in binding CD28 [34], and computational analyses evidenced the potential high-affinity binding of the motif enclosing the PRRARS insert of the SARS-CoV-2 spike to both TCR and CD28 [19]. These data, together with the evidence that MIS-C were associated with a skewed TCRVβ repertoire [19,20,21,22], suggested that the hyperinflammation observed in MIS-C could be due to the SAg-like activity of the SARS-CoV-2 spike. However, no functional data on the capability of the SARS-CoV-2 spike to stimulate inflammatory cytokine production in T cells have been reported.

Here, we examined the intrinsic SAg-like activity of the SARS-CoV-2 spike by comparing the production of inflammatory cytokines in both the TCRVβ3^+^ Jurkat T cell line and peripheral blood CD4^+^ T cells stimulated with the SARS-CoV-2 spike or staphylococcal SEB. Our data show that, in contrast to SEB, the SARS-CoV-2 spike does not elicit inflammatory cytokine production in T cells, evidencing the lack of intrinsic SAg-like activities in the SARS-CoV-2 spike.

## 2. Materials and Methods

### 2.1. Cells, Abs, and Reagents

Human primary CD4^+^ and CD8^+^ T cells were purified from the buffy coats of anonymous healthy donors (HD) (Policlinico Umberto I, Sapienza University of Rome, Italy) by negative selection using magnetic isolation kits (#17952, STEMCELL Technology) and positive selection using an MACS microbeads sorting kit (#130-045-201, Miltenyi Biotec, Milan, Italy), respectively, and cultured in RPMI 1640 supplemented with 5% human serum (Euroclone, UK), L-glutamine, penicillin, and streptomycin. Written informed consent was obtained from all HD, and the entire procedure was approved by the Ethics Committee of Policlinico Umberto I (Ethical code N. 1061bis/2019, 13 September 2019). The purity of the sorted population was 95–99%, as evidenced by staining with anti-CD3 plus anti-CD4 or anti-CD8 Abs.

The CD28^+^ CH7C17 Jurkat T cell line expressing TCR Vβ3.1 [47,48] was cultured as previously described [41]. Murine L cells (5-3.1/B7) co-transfected with HLA-DRB1*0101 and B7.1/CD80 [49,50] and HeLa cells (ATCC number: CCL2™) stably transfected with HLA-A*0201 were used as APCs [51,52].

The following antibodies were used: anti-human CD80-FITC (#21270803, 1:10 dilution), anti-human CD4-FITC (#21278043, 1:10 dilution), anti-human CD86-PE (#555658, 1:10 dilution) (ImmunoTools, Germany); anti-human CD28-PE (# 130-109-441, 1:10 dilution), anti-human CD4-PE (#130-091-231, 1:10 dilution) (Miltenyi Biotec, Italy); mouse anti-human CD28 (CD28.2, #555726, 2 μg mL^−1^), mouse anti-human CD3 (UCHT1, #555330, 2 μg mL^−1^), goat anti-mouse (GAM, # 553998, 2 μg mL^−1^), anti-human CD3-PE (#555333, 1:10 dilution) (BD Biosciences, Milano, Italy); anti-human HLA-DR-PE (#12-9956-71, 1:10 dilution) (ThermoFisher Scientific, Rome, Italy).

Staphylococcal Enterotoxin B (SEB, #54881) and recombinant SARS-CoV-2 spike protein (S1/S2) containing a C-terminal His-Tag (aa14-1213, #RP-87668) were purchased by Merck (Italy) and Invitrogen (ThermoFisher Scientific), respectively.

### 2.2. Cytokine Production (ELISA)

Then, 2 × 10^6^ mL^−1^ Jurkat T cells and primary CD4^+^ or CD8^+^ T cells were plated in a flat-bottomed 24-well plate and either unstimulated or stimulated for 24 h with 2 μg mL^−1^ of crosslinked anti-CD3 (UCHT1) or anti-CD3 plus anti-CD28 (CD28.2) antibodies (Abs) or anti-CD3 Abs plus 1 μg mL^−1^ of the SARS-CoV-2 spike or 1 μg mL^−1^ of the SEB or SARS-CoV-2 spike in the presence or absence of adherent 5-3.1/B7 cells or HLA-A*0201-expressing HeLa cells. In some experiments, the SARS-CoV-2 spike was heat denatured by water boiling (100 °C) for 10 min. At this temperature, most of the proteins lose their native contacts and secondary structure [53]. The secretion of human inflammatory cytokines in the culture supernatants was measured by using human IL-8 (#DY208-05), IL-2 (#DY202-05), TNF-α (#DY210-05), IFN-γ (#DY285B-05), and IL-6 (#DY206-05) ELISA kits, according to the manufacturer’s instructions (R&D Systems). The experiments were performed in duplicate, and the data were analysed on a Bio-Plex (Bio-Rad, Hercules, CA, USA). The sensitivity of the assays was 9.4 pg mL^−1^ for IL-6 and IFN-γ, 15.6 pg mL^−1^ for IL-2 and TNF-α, and 31.3 pg mL^−1^ for IL-8.

### 2.3. Plasmids, Cell Transfection and Luciferase Assays

The following luciferase reporter constructs were used: NF-κB luciferase gene under the control of six thymidine kinase NF-κB sites [54], NF-AT luciferase reporter construct containing the luciferase gene under the control of the human IL-2 promoter NF-AT binding site [55], and the AP-1-luciferase construct containing the luciferase gene under the control of two human collagenase TRE sites [56].

Then, 10^7^ Jurkat cells were electroporated at 260 V, 960 µF in 0.5 mL RPMI 1640 medium supplemented with 10% FBS with 2 μg of NF-κB-luciferase or 10 μg of NF-AT-luciferase or the AP-1 luciferase constructs together with 5 μg of the pEGFP construct, keeping the total amount of DNA constant (30 µg) with an empty vector. After 24 h, the cells were stimulated with 5-3.1/B7 cells in the presence or absence of SEB or the SARS-CoV2 spike (1 μg mL^−1^) at 37 °C for 6 h. The luciferase activity was measured according to the manufacturer’s instructions (Promega). The luciferase activity, measured in triplicates, was expressed as fold inductions (F.I.) after the normalisation to GFP values.

### 2.4. SDS-PAGE

The recombinant SARS-CoV-2 spike protein and SEB were resolved by 10% or 12% SDS-PAGE, respectively. The gels were fixed and stained with Coomassie blue (50% ethanol, 10% acetic acid, 0.1% Coomassie blue).

### 2.5. Structural Modeling of the Putative Interactions between SARS-CoV-2 Spike and TCR as Well as between SEB and 6D3 Ab or CD28

The structure predictions were performed in a standalone platform of AlphaFold 2 and AlphaFold-Multimer [57], as implemented in ColabFold, which was set up on a local computer with a Linux operating system and accelerated with two NVIDIA GeForce RTX 2080 Ti GPU. The “Template mode” using PDB 7N1Q [58] (Sequence Identity over aligned regions: 99.5%) was used for this purpose. Protein–Protein Docking was performed with ClusPro 2.0 with Immunoglobulin (Ig)-like structures docking [59], using as an input the previously obtained model of the Spike and the αβTCR structure (PDB: 2XN9), SEB and 6D3 Ab (PDB: 4RGN) and CD28 (PDB: 1YJD). The search was constrained to include the complementary determining regions (CDRs) at the interface of interaction. The other parameters were kept at their default values.

### 2.6. Statistical Analysis

The sample size was chosen based on previous studies to ensure adequate power. Parametrical statistical analysis (mean and SEM) was performed to evaluate the differences between continuous variables through Prism 8.0 (GraphPad Software, San Diego, CA, USA) by one-way ANOVA or Mann–Whitney tests. For all of the tests, *p* values < 0.05 were considered statistically significant.

## 3. Results

### 3.1. Analysis of SAg-like Inflammatory Activity of SARS-CoV-2 Spike Protein

Recent computational simulations generated on the basis of the cryoelectron microscopy of the SARS-CoV-2 spike [60] and the X-ray structure of human TCR TRAV27/TRBV19 in a ternary complex with HLA-DR1 and the staphyloccoccal SAg, SEH [61], suggested a unique putative binding site for the TCR Vβ chain near the S1/S2 cleavage site of the SARS-CoV-2 spike [19]. Further examination of the TCR Vβ-binding region on the SARS-CoV-2 spike (T_678_NSPRRARSVASQ_690_) also suggested strong structural similarities to the superantigenic fragment of staphylococcal SEB that has been recently identified to bind to CD28 costimulatory molecules, thus triggering an inflammatory cytokine storm [34,37,38,62]. However, the functional relevance of this putative SAg activity of the SARS-CoV-2 spike is presently unknown [63]. Here, we analysed the inflammatory activity of the SARS-CoV-2 spike on T cells in comparison with staphylococcal SEB. To this end, we used a recombinant SARS-CoV-2 spike protein covering S1 and S2 subunits (amino acids 14-1213) with an intact SAg-like TNSPRRARSVASQSA sequence produced in *E. coli* and containing a C-terminal His-tag. The usage of *E. coli* is often not recommended for challenging proteins that undergo glycosylation and require complex folding, such as the SARS-CoV-2 spike [64,65]. However, recent structural data from the circular dichroism and gel-filtration chromatography of the His-tagged recombinant SARS-CoV-2 spike proteins produced in *E. coli* and in mammalian HEK-293 cells evidenced that both of the proteins are stable and correctly folded, and capable of binding angiotensin-converting enzyme 2 (ACE2) [66]. Moreover, the functional data provided by ThermoFisher R&D also demonstrated that the *E. coli* recombinant SARS-CoV-2 spike (#RP-87668) efficiently recognised and bound ACE2 (proprietary data). After verifying the purity of the SARS-CoV-2 spike and SEB by SDS-PAGE (Appendix A), we firstly analysed the SAg activity of the SARS-CoV-2 spike on a CD28^+^ CH7C17 Jurkat T cell line expressing TCRVβ3.1 [47,48] that specifically interacts with SEB [67] and potentially with the putative CD28-binding site of the SARS-CoV-2 spike [19]. Although we have recently demonstrated that MHC-II molecules are dispensable for SEB inflammatory activity [38], the models and simulations of the SAg activity of the SARS-CoV-2 Spike have been conducted on a trimolecular complex involving HLA-DR1 [19,61]. Accordingly, we used murine 5-3.1/B7 cells co-expressing human HLA-DRB1*0101 and B7.1/CD80 molecules as APCs, which efficiently bind to SAgs and activate Jurkat and primary T cells without processing and presenting the derived peptides in association to MHC-II [38,47,68]. The stimulation of CD28^+^ TCRVβ3.1^+^ Jurkat cells with SEB alone induced significant TNF-*α* production (mean = 114 pg mL^−1^) that further increased by at least six-fold (mean = 697 pg mL^−1^) in the presence of MHC-II/B7-expressing 5-3.1/B7 cells (Figure 1a). On the contrary, IL-2 secretion was observed only when the Jurkat cells were stimulated with SEB in the presence of 5-3.1/B7 cells (Figure 1b). Consistent with our previous data [38,41], Jurkat cell stimulation with B7-expressing APCs induced the up-regulation of IL-8 secretion (mean = 276 pg mL^−1^) that strongly increased in the presence of SEB (mean = 3395 pg mL^−1^) (Figure 1c). In contrast to SEB, the stimulation of Jurkat cells with equal amounts of SARS-CoV-2 spike did not induce any significant production of TNF-*α* (Figure 1a) and IL-2 (Figure 1b) neither alone nor in the presence of 5-3.1/B7 cells. A slight increase in IL-8 secretion was observed when the Jurkat cells were stimulated with the SARS-CoV-2 spike in the presence of 5.3-1/B7 cells (mean ± SEM: 377.6 ± 76.63), but no significant differences were detected when compared to the stimulation with 5-3.1/B7 cells alone (mean ± SEM: 276.5 ± 18.5) (Figure 1c). Similar results were obtained by analysing the three major transcription factors regulating inflammatory cytokine expression, NF-AT, AP-1, and NF-*κ*B. The stimulation of Jurkat cells with SEB alone was able to induce a significant up-regulation of NF-AT (Figure 1d), AP-1 (Figure 1e), as well as NF-*κ*B luciferase activities (Figure 1f), which were strongly enhanced in the presence of 5-3.1/B7 cells (Figure 1d–f). On the contrary, no significant activation of NF-AT, AP-1, and NF-*κ*B was induced by the SARS-CoV-2 spike neither alone nor in the presence of 5-3.1/B7 cells. Consistent with the ability of CD28 stimulation alone to induce NF-*κ*B activation [42], the stimulation of Jurkat cells with B7.1-expressing 5-3.1/B7 cells induced a significant up-regulation of NF-*κ*B luciferase activity (mean ± SEM: 6.58 ± 0.86) that was significantly increased by SEB (mean ± SEM: 9.75 ± 0.85) but not by SARS-CoV-2 spike (mean ± SEM: 4.92 ± 0.63) (Figure 1f). Therefore, despite the SAg signature suggested by the in silico models, the SARS-CoV-2 spike is not able to stimulate inflammatory responses in Jurkat T cells expressing CD28 and a SEB-binding TCRVβ chain.

Since Vβ skewing in children with MIS-C involves TCRVβ21.3 (TRBV11-2), 24.1, and 11.3 [20,21,22,69], which are not signature targets of SEB [70], we next analysed the SAg-activity of the SARS-CoV-2 spike T cells isolated from the peripheral blood of healthy donors (HD), which express all TCR Vβ-families, including those amplified in MIS-C [71]. As we have previously observed [38], peripheral blood CD4^+^ T cells from HD expressed high levels of CD28 (mean = 92) but very low levels of HLA-DR (mean = 3.4), B7.1/CD80 (mean = 1) and B7.2/CD86 (mean = 6.8) molecules (Figure 2a,b). The stimulation of CD4^+^ T cells with SEB alone induced the significant production of TNF-α (mean = 1393 pg mL^−1^), IL-6 (mean = 456 pg mL^−1^), IL-2 (mean = 1467 pg mL^−1^), and IFN-γ (mean = 258 pg mL^−1^), which strongly increased in the presence of 5-3.1/B7 cells (Figure 2c–f). Conversely, the SARS-CoV-2 spike only induced a mild increase in TNF-α (mean = 117 pg mL^−1^) and IL-6 (mean = 256 pg mL^−1^) that did not further increase to the massive levels induced by SEB when 5-3.1/B7 cells were added to the culture (TNF-α, SEB = 8907, spike = 112; IL-6, SEB = 946, spike = 342) (Figure 2c,d). No significant IFN-γ production was detected following the stimulation of CD4^+^ T cells with the SARS-CoV-2 spike neither alone nor in the presence of 5-3.1/B7 cells (Figure 2f) and very low levels of IL-2 (mean = 36 pg mL^−1^) were detected when the T cells were stimulated with the SARS-CoV-2 spike in the presence of 5-3.1/B7 cells (Figure 2e). To exclude the possibility that SARS-CoV-2 spike folding could interfere with the exposure of the TNSPRRAR SAg-like motif and the putative TCR/CD28 binding as well as the activation of CD4^+^ T cells [19], we also stimulated T cells with a heat-denatured SARS-CoV-2 spike (boiled) and we did not detect any significant change in the secretion of inflammatory cytokines compared to the native form (Figure 2g–j). Since the superantigen-like insert of the SARS-CoV-2 spike has been described to bind to CD28 costimulatory molecules [19], we tested the potential costimulatory activity of the SARS-CoV-2 spike to trigger second-order superantigenic responses [63] in anti-CD3 stimulated T cells. Consistent with previous data [41,44], a significant production of TNF-α (Figure 3a, mean = 1699 pg mL^−1^) and IL-2 (Figure 3b, mean = 638 pg mL^−1^) was only induced when CD3 and CD28 were co-engaged. CD3 stimulation also induced an IFN-γ production (Figure 3c, mean = 4403 pg mL^−1^) that significantly increased following CD28 engagement (mean ± SEM = 7282 pg mL^−1^). On the contrary, no significant increase in TNF-α (Figure 3a), IL-2 (Figure 3b), and IFN-γ (Figure 3c) was observed when the CD4^+^ T cells were stimulated with anti-CD3 Abs in the presence of the SARS-CoV-2 spike.

Finally, since the expansion of TCRVβ21.3 in MIS-C patients has been recently associated with three HLA class I alleles (A02, B35 and C04), we compared the SAg activity of SEB and the SARS-CoV-2 spike in CD4^+^ or CD8^+^ T cells stimulated with 5-3.1/B7 cells or HeLa cells stably transfected with the HLA-A*0201 allele [52]. The stimulation of CD4^+^ T cells with SEB in the presence of HeLa cells (Figure 4a,c,e) induced a high production of TNF-*α* (mean = 1886 pg mL^−1^), IL-2 (mean = 1557 pg mL^−1^), and IFN-*γ* (mean = 1846 pg mL^−1^), although significantly lower than that induced by 5-3.1/B7 cells (TNF-*α* mean = 5915 pg mL^−1^; IL-2 mean = 5377 pg mL^−1^; IFN-*γ* mean = 3501). A significant up-regulation of IL-2 (Figure 4c) and IFN-*γ* (Figure 4e) was also observed in the CD8^+^ T cells stimulated with SEB in the presence of both 5-3.1/B7 cells (IL-2 mean = 1103; IFN-*γ* mean = 655.7) and HeLa cells (IL-2 mean = 1088; IFN-*γ* mean = 653). On the contrary, no significant TNF-*α* (Figure 4b), IL-2 (Figure 4d), or IFN-*γ* (Figure 4f) were detected following the stimulation of CD4^+^ or CD8^+^ T cells with the SARS-CoV-2 spike in the presence of HeLa cells.

Altogether, these in vitro data show that SARS-CoV-2 spike does not possess an intrinsic canonical SAg activity targeting TCR and/or CD28 that may explain the hyperinflammation observed in severe MIS-C.

### 3.2. Structural Reassessment of the Putative Interaction between SARS-CoV-2 Spike and TCR

The discrepancy between our functional data on the lack of SAg-like activity of the SARS-CoV-2 spike and the computational models of the putative interaction between the SARS-CoV-2 spike and TCRVβ [19] prompted us to repeat the computational simulations with recent state-of-the-art tools, which were not available at the time of the study [19]. We relied on the recently released artificial intelligence (AI)-based AlphaFold2 algorithm [57], which is known to impressively outperform any other protein structure prediction tool, to model the SARS-CoV-2 spike protein. AlphaFold2 also took advantage of the structural information from the recently released structures of the spike at a high resolution (<3.0 Å, e.g., PDB 7N1Q), which were not previously available [58]. This approach provided us with a much more accurate prediction of the SARS-CoV-2 spike region 661–685. Therefore, by comparing such a region with the structure of the SEB superantigenic fragment (150–161), we obtained an RMSD of ~2.0 Å, which is a relatively high value considering the small extension of the region superposed. Moreover, the length and secondary structure composition of the two regions are completely different, with the SEB superantigenic fragment 150–161 folding into a helix–strand–loop motif, while the spike region 661–685 folded into a loop–hairpin motif (Figure 5a). The SEB superantigenic fragment is predicted to be a conformational binding epitope of both neutralising 6D3 Ab (Appendix A) [72] and CD28 (Appendix A) [73], roughly corresponding to the 800 Å2 solvent-accessible area buried upon complex formation. The conservation of the β-strand(8)/hinge/*α*-helix(4) of the SEB superantigenic fragment, together with the whole three-dimensional fold of SEB, is pivotal for CD28 binding [34]. By losing this helix–strand–loop motif, the binding of the SARS-CoV-2 spike region 661–685 to CD28 is unlikely.

We next repeated the protein–protein docking approach, using the same previously described input data [19] and a recently developed mode of ClusPro 2.0 for use with Immunoglobulin (Ig)-like structure docking [59]. This method is known to produce much better results compared to the ClusPro 2.0 standard approach due to the use of the new potentials significantly improving the performance of docking for Ig–protein antigen complexes, and the search was constrained to include the complementary determining regions (CDRs) at the interface of the interaction [59]. The obtained results led to 24 different clusters, with the top-ranking one populated by 110 members with a mean energy of −267.4. The top-ranking model that we found predicts an interaction between the human TCR TRAV27/TRBV19 [61] and an epitope encompassing residues 623–636 of the SARS-CoV-2 spike (Figure 5b, Appendix A) that is located near but not overlapping, the PRRARS insert (residues 661–685). In only one case (Cluster 6; 20 members; mean energy of −283.2), the spike-binding epitope contained the PRRARS insert (Appendix A). Most importantly, as shown in Appendix A, no predicted model outperformed the other models in terms of interaction energy and surface complementarity, evidencing again the extreme difficulty of properly ranking the obtained docking results.

## 4. Discussion

The clinical similarity between MIS-C and TSS [21,24] has recently led to the hypothesis that an SAg-like element in SARS-CoV2 may cause the hyperinflammatory immune responses observed in some COVID-19-affected children [26,27]. Consistently, recent in silico modelling evidenced that the SARS-CoV-2 spike contains a PRRARS insertion near the S1/S2 furin cleavage site, highly conserved in all spike variants, including Omicron [46], within the motif, YQTQTNSPRRARS, which is structurally similar to the SAg fragment of SEB, TNKKKATVQELD, and potentially able to bind TCR and CD28 as well as MHC-II with high affinity and to elicit the massive production of inflammatory cytokines [19]. However, no in vitro data supporting the SAg-like activity of SARS-CoV-2 were provided. Herein, we addressed this issue by analysing the capability of the SARS-CoV2 spike to induce inflammatory cytokine production in both Jurak cells and primary CD4^+^ T cells, which are highly responsive to SEB stimulation. Our in vitro data evidence that, in contrast to SEB, the SARS-CoV2 spike does not exhibit an SAg-like activity targeting TCR and/or CD28, thus suggesting that the immunopathological mechanisms leading to MIS-C are distinct from SEB-induced TSS.

SEB-mediated TSS relies on the ability of SEB to induce the polyclonal activation of T cells by binding with high-affinity specific TCRVβ subsets such as Vβ3, 12, 13.2, 14, 17, and 20 [70], the HLA-DR isotype of MHC-II molecules [74] and the CD28/B7 costimulatory axis [34,36,38,62,75,76]. The three-dimensional structure of the SEB/TCR binary complex and the SEB/TCR/MHC-II ternary complex [28,77] highlighted that the SEB regions involved in binding TCR and MHC-II are distal from the CD28 binding site [76]. In particular, SEB interacts with TCR through a TCR-binding cleft involving 14 residues from SEB (T_18_G_19_L_20_E_22_N_23_V_26_L_58_N_50_T_90_T_91_R_110_F_177_N_178_Q_210_), that together with 17 residues from the TCRVβ chain, create the TCR–SEB interface [77]. The X-ray crystallographic structure of SEB in a complex with HLA-DR1 evidenced that SEB binds to MHC-II outside of the peptide-binding domains, and the SEB-MHC-II interface involves SEB residues 33–39, 48–52, and 63–68 [28,77]. Similar domains have not been identified in the SARS-CoV-2 spike that only contains an SAg-like motif structurally similar to the 12 amino-acid b-strand (8)/hinge/a-helix (4) of SEB involved in binding the homodimer interface of CD28 but neither TCR nor MHC-II [34,37,62,76]. Consistently, the SEB stimulation of the TCRVβ 3.1^+^ Jurkat T cell line with APCs expressing both HLA-DR1 and B7.1 elicited a massive production of inflammatory cytokines (Figure 1). Similar results were obtained from the peripheral blood CD4^+^ T cells from HD (Figure 2), which express all of the TCRVβ chain families [71]. In contrast to SEB, the SARS-CoV-2 spike failed to induce the activation of inflammatory pathways and no massive secretion of inflammatory cytokines in either Jurkat cells (Figure 1) or CD4^+^ T cells (Figure 2) or activated CD4^+^ T cells (Figure 3) was detected. The analysis of TCRVβ skewing in children with severe MIS-C showed an enrichment of TCRVβ21.3 (TRBV11.2), 24.1, and 11.3 [20,21,22,69], which are not signature targets of SEB [70]. In particular, TCRVβ21.3 expansion in MIS-C was found to be associated with HLA class I alleles [22]. The lack of SARS-CoV-2 spike inflammatory activity in both CD4^+^ and CD8^+^ T cells stimulated with HLA-A*0201-expressing HeLa cells (Figure 4) evidenced the inability of the putative SAg-like fragment in the SARS-CoV-2 spike to activate T cell inflammatory responses in vitro.

In silico protein–protein docking suffers from several problems, e.g., modelling the physics of the system, solvent effects, dynamics, and the difficulty in accurately ranking the docked results. Hence, the hypotheses generated from the in silico approaches should always be validated to ensure that these algorithms continue to improve in their accuracy and usefulness. For instance, by using advanced computational tools (AlphaFold2, ClusPro 2.0 “Antibody mode”), which were not available at the time of the study, by Cheng et al. [19], we provide strong evidence that the alleged sequence and structural similarities between SARS-CoV-2 spike region 661–685 and SEB are weak (Figure 5a), and the SARS-CoV-2 spike lacks the β-strand(8)/hinge/α-helix(4) secondary structure that is pivotal for the binding of SEB to CD28 [34] (Appendix A). Moreover, we also evidence that the putative SAg-like motif within the SARS-CoV-2 spike (the PRRARS insert) is not involved in TCRVβ binding (Figure 5b and Appendix A). These data emphasise the importance of the functional validation of the assumptions drawn in silico.

In contrast to TSS, inflammatory cytokine production in MIS-C occurs several weeks after infection and, in many patients, it peaks when the SARS-CoV-2 virus is no more detected and no specific viral or bacterial signatures can be evidenced [12,13,21,23,69,78]. In a few studies, the majority of MIS-C patients were positive for SARS-CoV-2 RNA in either the nasopharyngeal tract [79] or the gastrointestinal tract, where it has been suggested to favour the release and persistence of the SARS-CoV-2 spike and S1 proteins in the plasma of children with MIS-C [80]. More recent data obtained by using an artificial intelligence (AI)-based approach also revealed that the host immune responses in MIS-C and in Kawasaki disease (KD) are similar and share an IL-15/IL-15RA pathway [78] that is common to the viral pandemic (ViP) [81]. This ViP signature has been demonstrated to depend on the interaction of the SARS-CoV-2 spike with ACE2 rather than with the TCRVβ [81]. Moreover, a strong autoimmune signature of MIS-C has been reported in all studies [18,20,21,69], characterised by the clonal expansion of short-lived plasmablasts [69,79] and by the production of high levels of autoantibodies [18,20,69,82], which may lead to the generation of immune complexes that, by activating the classical complement pathway, may cause both tissue injury and hyperinflammation [20].

In conclusion, our data show the lack of intrinsic inflammatory activity by the SARS-CoV-2 spike, suggesting that the pathogenesis of MIS-C is distinct from SAg-mediated TSS. Instead, MIS-C pathogenesis could be related to the aberrant adaptive immune response against SARS-CoV-2 infection that may lead, in rare cases and in genetically predisposed children, to transient and cross-reactive autoimmune responses. Consistently, a two-hit model for MIS-C has been recently proposed whereby the first hit triggered by SARS-CoV-2 infection is followed by a second microbial infection that, in turn, drives hyperinflammation and the expansion of T and B cell subpopulations, which are cross-reactive against self-antigens [27,69]. Alternatively, the high anti-SARS-CoV-2 IgGs produced in children with severe MIS-C [82,83] may, in turn, hyperactivate monocytes through their FcγR, thus contributing to systemic inflammation [84].

## Figures and Tables

**Figure 1 cells-11-02526-f001:**
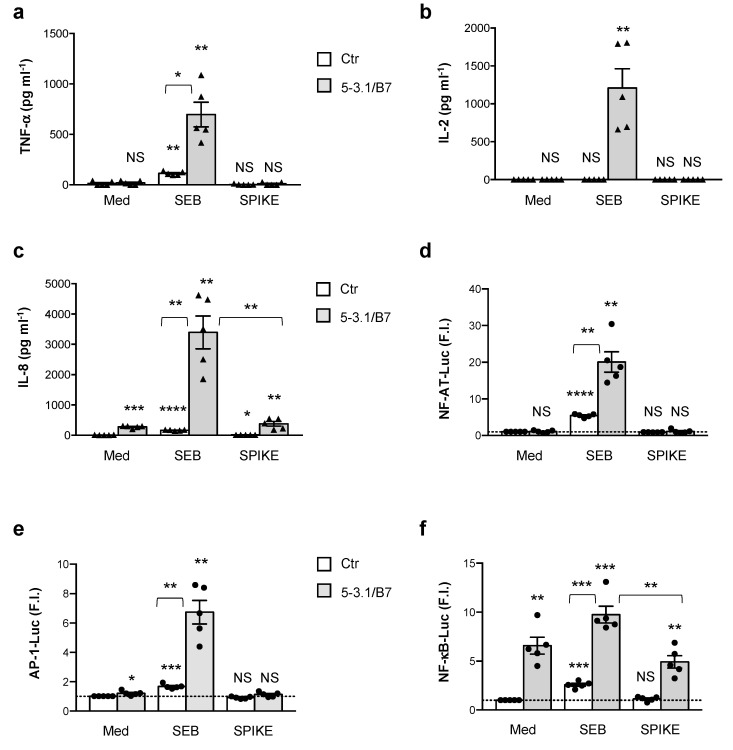
SARS-CoV-2 spike does not stimulate inflammatory pathways in Jurkat cells expressing CD28 and TCRVb3.1. (**a**–**c**) CH7C17 Jurkat cells expressing CD28 and TCRVβ3.1 (n = 5) were cultured for 24 h with medium alone (Med) or 1 μg mL^−1^ of SEB or 1 μg mL^−1^ SARS-CoV-2 spike in the absence (Ctr) or presence of 5-3.1/B7 cells expressing human HLA-DR1 and B7.1/CD80 molecules. TNF-α (**a**), IL-2 (**b**) and IL-8 (**c**) secretion in culture supernatants was measured by ELISA. Data show the mean ± SEM, and statistical significance was calculated by one-way ANOVA. (**d**–**f**) Jurkat cells (n = 5) were transfected with 5 μg GFP together with 10 μg NF-AT-luciferase (Luc) (**d**), or 10 μg AP-1-Luc (**e**) or 2 μg NF-κB-Luc (**f**) constructs and then unstimulated (Ctr) or stimulated for 6 h with SEB or SARS-CoV-2 spike alone or in the presence of 5-3.1/B7 cells. The results were expressed as fold inductions (F.I.) over the basal level of luciferase activity in unstimulated cells after normalisation to GFP values. Bars show mean ± SEM of five independent experiments. Statistical significance was calculated by comparing each group with unstimulated cells (Ctr) cultured with medium alone (med) or between conditions indicated by the arcs by one-way ANOVA. (*) *p* < 0.05, (**) *p* < 0.01, (***) *p* < 0.001, (****) *p* < 0.0001. NS = not significant.

**Figure 2 cells-11-02526-f002:**
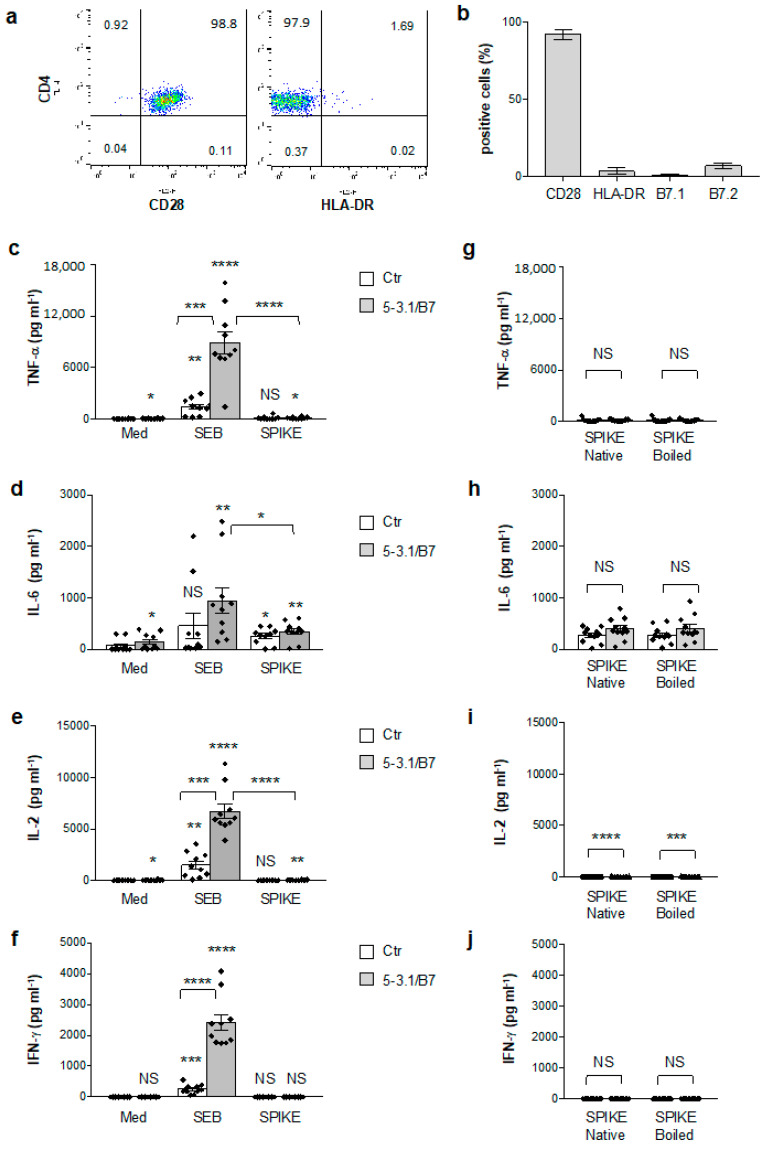
SARS-CoV-2 spike does not stimulate inflammatory cytokine production in peripheral blood CD4^+^ T cells. (**a**) Representative FACS analysis of human CD4^+^ T cells isolated from the peripheral blood of HD stained with anti-CD4-FITC plus anti-CD28-PE or anti-HLA-DR-PE. (**b**) The percentage of human CD4^+^ T cells from HD (n = 5) expressing CD28, HLA-DR, B7.1/CD80 or B7.2/CD86 was calculated. The results express the mean percentage of positive cells ± SEM. (**c**–**f**) Peripheral blood CD4^+^ T cells from HD (n = 10) were cultured for 24 h with medium alone (Med) or SEB or SARS-CoV-2 spike in the absence (Ctr) or presence of 5-3.1/B7 cells. TNF-α (**c**), IL-6 (**d**), IL-2 (**e**) and IFN-γ (**f**) secretion in culture supernatant was measured by ELISA. Data show the mean ± SEM. Statistical significance was calculated by comparing each condition with unstimulated cells (Ctr) cultured with medium alone (med) or between conditions indicated by the arcs by one-way ANOVA. (**g**–**j**) Peripheral blood CD4^+^ T cells from HD (n = 10) were stimulated for 24 h with native SARS-CoV-2 spike or heat-denatured SARS-CoV-2 spike (boiled) in the absence (Ctr) or presence of 5-3.1/B7 cells. TNF-α (**g**), IL-6 (**h**), IL-2 (**i**) and IFN-γ (**j**) levels were measured by ELISA. Data show the mean ± SEM. Statistical significance was calculated by Mann–Whitney test. (*) *p* < 0.05, (**) *p* < 0.01, (***) *p* < 0.001, (****) *p* < 0.0001. NS = not significant.

**Figure 3 cells-11-02526-f003:**
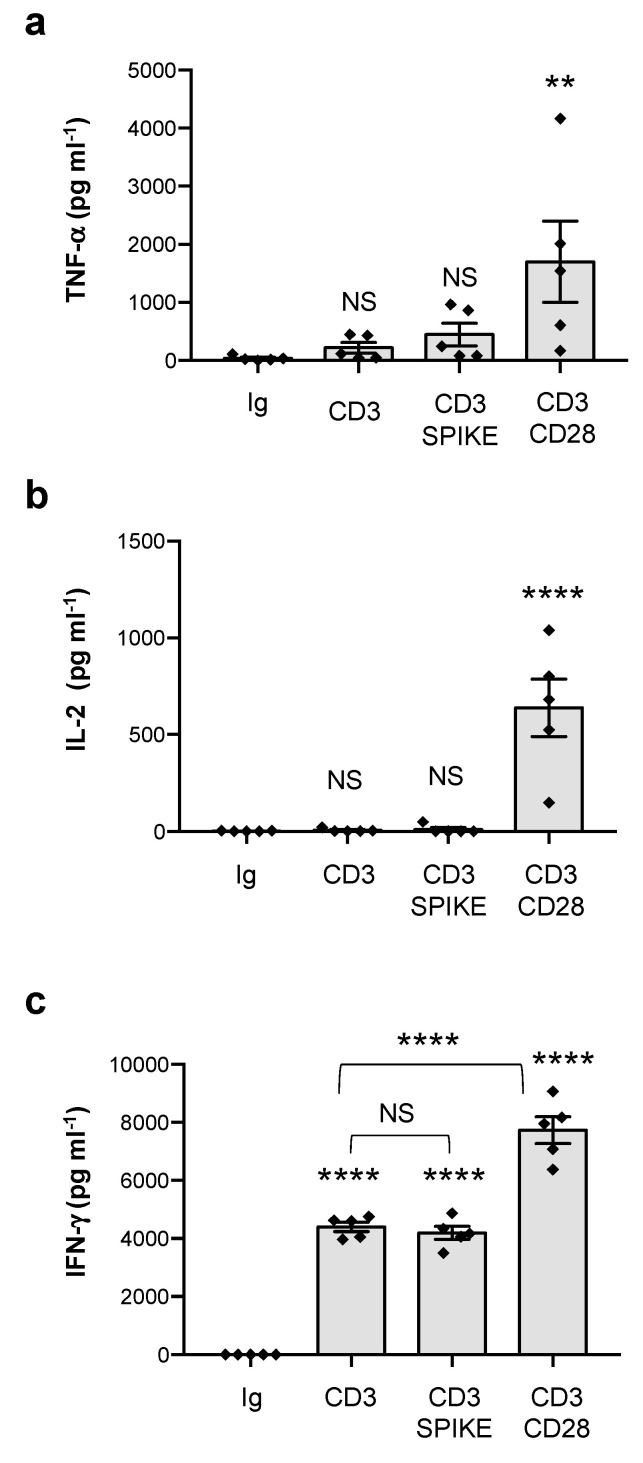
SARS-CoV-2 spike does not co-stimulate inflammatory cytokine production in anti-CD3 activated CD4^+^ T cells. (**a**–**c**) Peripheral blood CD4^+^ T cells from HD (n = 5) were cultured for 24 h with isotype control Abs (Ig) or 2 μg mL^−1^ crosslinked anti-CD3 Abs (UCHT1) or anti-CD3 plus anti-CD28 (CD28.2) Abs or anti-CD3 Abs plus SARS-CoV-2 spike. TNF-α (**a**), IL-2 (**b**) and IFN-γ (**c**) secretions in culture supernatant were measured by ELISA. Data show the mean ± SEM. Statistical significance was calculated by comparing each condition with isotype control Abs (Ig) or between conditions indicated by the arcs by one-way ANOVA. (**) *p* < 0.01, (****) *p* < 0.0001. NS = not significant.

**Figure 4 cells-11-02526-f004:**
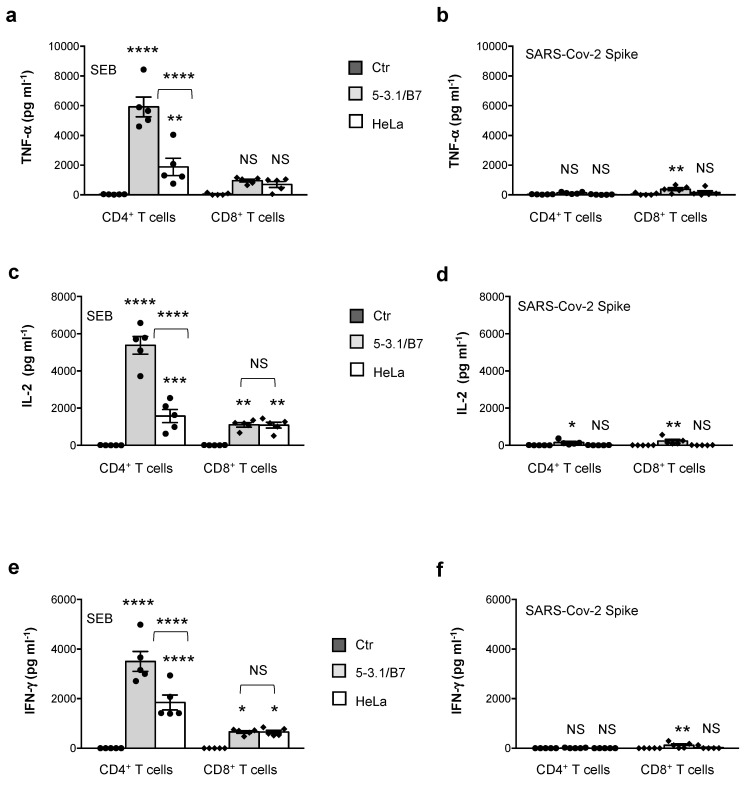
SARS-CoV-2 spike does not stimulate inflammatory cytokine production in CD4^+^ or CD8^+^ T cells stimulated with HeLa cells expressing HLA-A*0201. (**a**–**f**) Peripheral blood CD4^+^ T and CD8^+^ T cells from HD (n = 5) were cultured for 24 h with medium alone (Ctr) or SEB (**a**,**c**,**e**) or SARS-CoV-2 spike (**b**,**d**,**f**) in the presence of 5-3.1/B7 cells or HeLa cells expressing HLA-A*0201. TNF-α (**a**,**b**), IL-2 (**c**,**d**) and IFN-γ (**e**,**f**) secretion in culture supernatant was measured by ELISA. Data show the mean ± SEM. Statistical significance was calculated by comparing each condition with unstimulated cells cultured with medium alone (Ctr) or between conditions indicated by the arcs by one-way ANOVA. (*) *p* < 0.05, (**) *p* < 0.01, (***) *p* < 0.001, (****) *p* < 0.0001. NS = not significant.

**Figure 5 cells-11-02526-f005:**
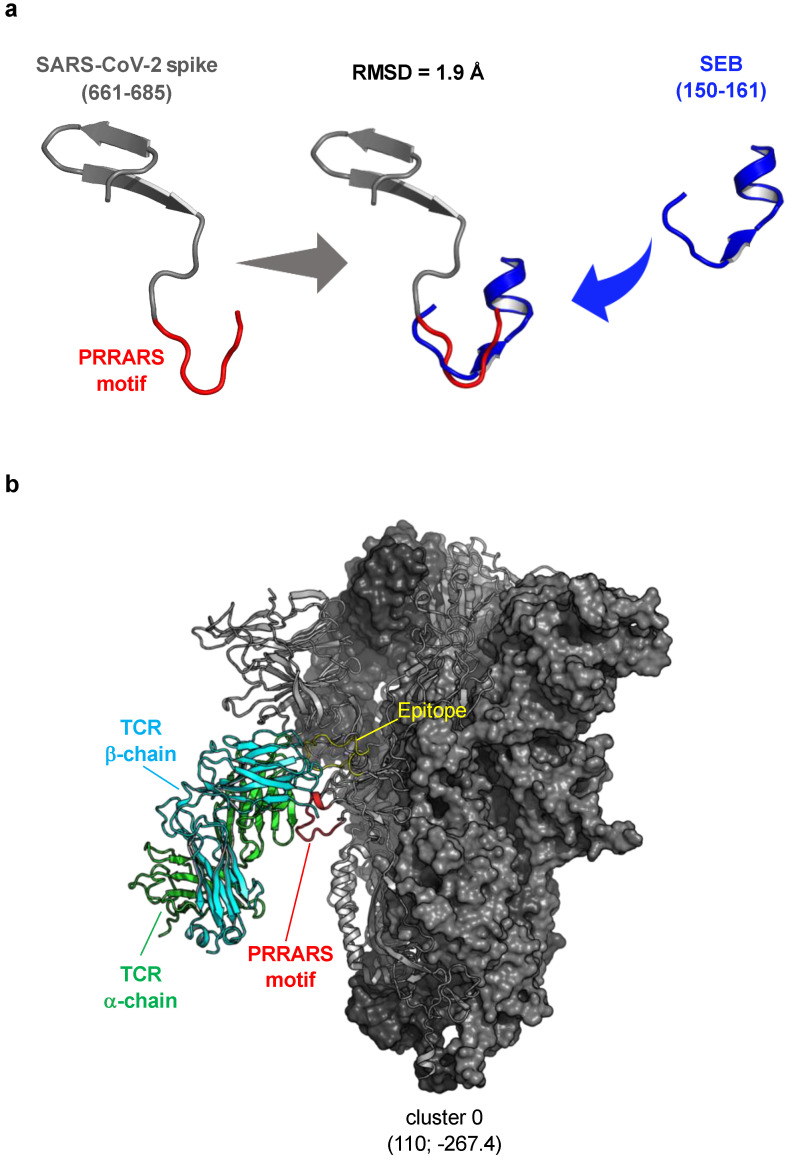
Structural prediction of the putative interaction between SARS-CoV-2 spike and TCR. (**a**) Structure superposition of SEB superantigenic fragment (150–161; blue) and the sequence of SARS-CoV-2 spike near the PRRARS insert (661–685; dark grey and red). (**b**) Top-1 cluster representative of SARS-CoV-2 spike and the human TRAV27/TRBV19 [61] complex, as predicted by ClusPro 2.0. The SARS-CoV-2 spike trimer is shown in grey, with the interacting chain as cartoons and the others as surface. The αβTCR structure (PDB: 2XN9) is represented as green (TCR α-chain) and cyan (TCR β-chain) cartoons. The PRRARS insert of the spike protein is coloured in red. The epitope that is recognised is coloured yellow. The number of representative structures within the cluster and the mean energy is shown in parenthesis.

## Data Availability

The data presented in this study are available on request from the corresponding author.

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
