# Peer review of "SARS-CoV-2 Spike Does Not Possess Intrinsic Superantigen-like Inflammatory Activity"

_cells, 2022, doi:10.3390/cells11162526_

Round 1

Reviewer 1 Report

The authors have incorporated all my suggestions and comments - thank you for doing extra experiments and including them in the manuscript. I have no additional comments.

Reviewer 2 Report

The manuscript "SARS-CoV-2 spike does not possess intrinsic superantigen-like inflammatory activity" has highlighted the important aspect of SARS CoV-2 spike protein. All the queries raised by reviewers has been addressed. The manuscript is sound and has broader reader coverage. 

This manuscript is a resubmission of an earlier submission. The following is a list of the peer review reports and author responses from that submission.

Round 1

Reviewer 1 Report

Authors are advised for minor grammatical correction in the manuscript 

Reviewer 2 Report

The conclusion they reached in this paper still cannot be supported by their experimental data provided even in the revision as there are many remaining serious methodological problems that preclude such a conclusion. Particularly troublesome is the new Fig 4 which was added by the authors who attempted to repeat the computational stimulation with major flaws, making the manuscript even more problematic and much less convincing.

Major problems that are persisting or newly added major problems:  New critiques are in Blue 

Original critique and answer provided for clarity of new critiques 

Original Critique 1a: While the computational studies show similarity of the recently discovered SARS CoV2 SAg -like motif in Si spike  to SEB the authors assume that the SAg motif would interact with the same TRBV genes and HLA alleles that SEB does. This is incorrect. In Figure 1, Vbeta3+ expressing Jurkat T cells are used. This is because SEB, which is used as a control, binds to human Vbeta3. However, Vbeta skewing in MIS-C does not involve Vbeta3 (as mentioned specifically by the authors on line 262). Therefore, this is not an appropriate experiment to measure Vbeta-specific activation by SPIKE protein. They tested Jurkat cells that express TCR VB3.1.  Why not test TRBV11.2 (Vb21.3) expressing cells.

Response 1a: We used CD28-positiveVbeta3-expressing Jurkat cells as control, because in the computational simulation by Cheng et al. the putative PRRARS SAg motif of SARS-CoV-2 Spike was shown to bind CD28 in a SEB-similar manner (Fig. 3E in Cheng et al 2020, doi: 10.1073/pnas.2010722117). Moreover, in the same paper, to examine the capability of SARS-CoV-2 Spike to bind TCR, the authors generated a structural model of SARS-CoV-2 spike based on the cryoelectron microscopy (cryo-EM) structure of the full spike protein and the X-ray structure of the human abTCR TRAV27/TRBV19, specific to the staphylococcal SEB, in a ternary complex with HLA-DR1 (Saline 2010 doi: 10.1038/ncomms1117). Similarly to Vb3, Vb skewing in MIS-C does not involve TRBV19 (Vb19). The information have been added in the results section of the revised manuscript (page 4, lines 154-161). 

New Critique 1 - The authors have still not properly addressed the query regarding the Vbeta3 specific activation in Figure 1. They are of course correct that SEB is reported to target this Vbeta, but this is not the case for the skewing in MIS-C patients. Thus, reading out Vbeta3 activation for the recombinant SPIKE protein does not make any sense. All this does is show SEB seems to be working.

Unfortunately, Authors also did not respond the point that was asked initially. Jurkat cells are CD8+ T cells, and authors used this experimental setting which works for SEB. Indeed, SEA does not induce cytokines in this unique experimental system in authors previous paper (Front. Immunol. 12:723689) , does not mean that SEA does not have SAg activity. In addition, the NF-kB luciferase activity mixed with human HLA expressing mouse L cells and Jurkat VB3,1 is very high without SEB, SPIKE, indicates mouse fibroblast and human lymphoma mix culture has potential inflammatory effect (Fig 1f).  Thus, they should have included another control, non human HLA transfected mouse L cells in Fig 1.  Authors should have evaluated T cell proliferation for SAg activity instead of cytokine production. It is quite strange that Authors is judging SAg activity for unestablished new super antigen candidate without a standard method. 

There is also a big pitfall in their assay system. Authors state… “…which express all TCR Vbeta-families, including those amplified in MIS-C (70)” and used human T cells from HD PBMCs.”, and used isolated human T cells.  However, Ref 70 they cited actually demonstrated that PBMCs T cells do not express all TCRVb, and lacks TRBV11, and even  lacks TRBV19 that authors newly proposed in this manuscript. Therefore, appropriate condition is required for testing SAg of SPIKE while the experiment worked for SEB which has broad range of TRBV reporters. While literature indicates TRBV11.2, 5-6, 14, 24 are involved in SARS-CoV-2, if author’s computational analysis found VRAV27/TRBV19 to bind SARS-COV-2, why did authors investigated TCR Vb 3.1 in Fig 1, and why did not test VRAV27/TRBV19 expressing T cell system?  The revised manuscript does not logically flow at all. I totally agree that “ in silico approaches should be always validated to ensure that these algorithms continue to improve in their accuracy”, then why did authors proposed TRBV19 to bind to spike protein without validating?  Authors response conflicts with their own manuscript.

Response 1b- Moreover, in the revised manuscript, we thoroughly reassessed the hypotheses and conclusions drawn by Cheng et al., and we repeated the computational simulations with advanced computational tools (AlphaFold2, ClusPro 2.0 "Antibody mode"), which were not available at the time of the study by Cheng et al. We provide strong evidence that the alleged sequence and structural similarities between SARS-CoV-2 S region 661-685 and SEB (150-161), reported by Cheng et al., are weak, and the putative SAg- like motif within SARS-Cov-2 spike (the PRRARS insert) is not involved in TCRVb binding. These data have been reported in the new Figure 4b, Figure S2 and Table Si of the revised manuscript and discussed (page 6, lines 249-281). Moreover, Cheng et al. used protein-protein docking approaches to obtain computational models of a putative interaction between the Spike glycoprotein of SARS-CoV-2 and TCRs. In silico protein-protein docking suffers from several problems, e.g. modelling the physics of the system, solvent effects, dynamics, and the difficulty in accurately ranking the docked results. Hence, hypotheses generated from in silico approaches should be always validated to ensure that these algorithms continue to improve in their accuracy and usefulness.  Inductive conclusions with insufficient evidences coming from computational studies can put perplexity into the literature and waste costly resources by following up on false positives, identified in poorly validated in silico studies. Finally, as highlighted by the reviewer, since Vb skewing in children with MIS-C involves TCRVb21.3 (TRBV11-2), 24.1 and 11.3, which are not signature targets of SEB, we analysed  SAg-activity of SARS-CoV-2 spike in T cells isolated from the peripheral blood of healthy donors (HD), which express all TCR V(3-families, including those expanded in MIS-C (Ochsenreither et al 2008, doi: 10.1186/1479-5876-6-34).  

New Critique  1b-  The computational analysis presented under the subtitle “structural reassessment…” has several flaws as summarized below.  First the level of similarity based on sequence alone might be considered moderate, but together with structural alignment, the similarity is certainly not weak. On the contrary, it is very strong. The problem in the current ‘assessment’ is that the structural comparison included in the revised version (Fig 4a) is misleading. They just show the backbone, and it is not clear how they align the two structures. The superposition of the side chains is important, and rather than a contiguous sequence, and as shown in the PNAS paper the spatial superimposition of the two sequences was noteworthy.  Here the authors have missed those features. AlphaFold-Multimer (referred in the revision) is generally not able to predict the binding of antibodies.  Furthermore, there is no evidence that antibody docking mode using ClusPro on AlphaFold2-predicted structures is able to improve docking results (see https://doi.org/10.1101/2021.10.04.463034).  

As to docking simulations, it is true that docking simulations have limitations and yield multiple models, however, the current state-of-the-art may provide good estimates if the quality (energetics/scores) of the predicted poses are reasonable and they take part in populated clusters, which were the case in the PNAS paper. In the current manuscript it is not clear whether they have properly evaluated the docked conformers. In the PDB structure (PDB: 7NN1Q) they used,  the "PRRAR" region is not resolved yet. Several structural similarities (including side chain orientations) reported in PNAS 2020 lack in their structural modeling and thus they reported different docking results.  

For those reasons Figure 4 ( with its associated supplements) and the newly added section on structural assessment shouldbe completely removed.

Original Critique (6): However, the most crucial issue is that they used full length Spike (S1 and S2). The SAg like motif is much more exposed in the S1 spike portion ( and not in the full Spike) Plus, this recombinant protein is HA tag, which further interferes with any potential SAg like function. Almost all recombinant Spike proteins – particularly the S1 portion spike have to mutate the key PRRAR motif ( so that they can purified the recombinant protein due to its excessive reactivity) – and thus missing key parts of the SAg motif  even is S1 spike is used. This protein would not be in trimeric form, likely highly unstable and is expressed by E. coli expression system so glycosylation likely very different to mammalian expressed Spike. Thus, the Spike proteins available commercially , including the current one used in this manuscript cannot address at all the question asked.

Response 6 : Firstly, we would like to underline that the commercial recombinant SARS-CoV-2 Spike used in this work contains the intact TNSPRRARSVASQSA SAg-like motif and is not HA-tagged, but is His-tagged at the C-terminus. We agree with the reviewer that the usage of E. coli is often not recommended for challenging prot folding such as Sars-CoV-2 spike. However, recent structural data from circular dichroism and gel filtration chromatography of His-tagged recombinant SARS-CoV-2 spike proteins produced in E. coli and in mammalian HEK-293 cells evidenced that both proteins are stable and correctly folded, and binds ACE2 (Maffei et al 2021, doi: 10.3390/biom11121812). Moreover, functional data provided by Thermo Fisher R&D also demonstrated that E. coli recombinant SARS-Cov-2 spike (#RP-87668) efficiently recognized and bound ACE2 (Figure R1 for referee, proprietary data). The information have been added in the result section of the revised  manuscript (page 4, lines 174-168). Finally, in the structural model simulated to verify SARS-CoV-2 spike binding to the TCR Vb chain, as well as to CD28, the authors 114 used the cryo-EM structure of the full spike protein, not the Si subunit, where they identified the PRRARS motif that is exposed to 115 the exterior and binds the CDR3 within Vb chain as well as CD28 (Fig. 1 and 3 in Cheng et al 2020, doi: 10.1073/pnas.2010722117). 116 In the structural model of SARS-CoV-2 Si trimer generated in the absence of either TCR or CD28, the exposure of the putative 117 SAmotif is only accentuated (Fig. 2B in Cheng et al 2020, doi: 10.1073/pnas.2010722117).

New Critique 2  ( for above ):  This answer would be sufficient if authors investigated Spike protein to bind to ACE2 but not for the SAg motif.  The fact that the Spike they use binds to ACE2 does not mean that it can also bind to the TCRs. Also company “proprietary data” should not be relied on 

Original Critique ((7a): Unlike bacterial superantigens (i.e. SEB) which are secreted toxins, the SARS-Cov-2 S, a glycosylated membrane bound trimer, exists in many conformations and undergoes proteolytic cleavage during cell infection to release the S1 and S2 (fusion) trimers. The complex nature of the spike machinery is difficult to capture in in vitro systems using recombinant proteins.

Response 7a: In the model generated to demonstrate the similarity of the sequence and structure properties between SEB and the PRRARS motif of Spike (Fig. 3E in Cheng et al 2020, doi: 10.1073/pnas.2010722117), the authors present the structural alignment of CD28, the receptor binding SEB, onto TCRVb domain, without the other spike subunits, thus suggesting the putative adaptability of monomeric SAg site to accommodate spike—TCRVb or SEB—CD28 interactions.

New Critique 3 ( for above)  :  Authors did not really respond to what was asked.  Authors simply mentioned about the Spike Motif to bind to TCRVb and CD28, but completely ignored the biological event and enzymatic activity of SARS-CoV-2 spike protein. 

Critique 7b: Preparation of recombinant viral protein with the endogenous sequence (PRRAR) does not allow for correct folding and/or stable conformational state, hence the use of a stable trimeric recombinant spike upon mutating the key region that corresponds to the superantigen-like fragment in many studies.  Many of the commercially available recombinant S1 proteins also have a C-terminal His tag that is immediately proximal to the SAg-like motif and interferes with its activity

Response 7b: As specified above (see the response to concern 6), C-terminal His recombinant SARS-CoV-2 spike proteins produced in E. coli and in mammalian HEK-293 cells are equally stable and correctly folded, and binds ACE2 (Maffei et al 2021, doi:10.3390/biom11121812). Moreover, functional data also demonstrated that E. coli recombinant SARS-Cov-2 spike efficiently recognizes and binds ACE2. The presence of a C-terminal His Tag motif has been shown to not interfere with the folding, stability and ACE binding.  In our commercial recombinant full Spike protein (aa 14-1213), the C-terminal His tag is not immediately proximal to the SAg-like P682RRAR685 motif, but after aa 1213. Moreover, functional data provided by Thermo Fisher R&D, also demonstrated that E. coli recombinant SARS-Cov-2 spike (#RP-87668) efficiently recognized and bound ACE2 (Figure for referee, proprietary data). The information have been added in the result section of the reviced manuscript (page 4, lines 172-174). 

New Critique 4 ( for above):  Spike protein generated in E. coli and HEK-293 cells would be equally stable.  But these were not tested and compared the functions beside ACE2 binding. 

The problem of making the SAg motif in E Coli and the C terminal His tag persists. And just because the commercial recombinant spike they used from Thermo Fisher R&D recognized and bound the ACE2 (proprietary data) doesn’t mean the peptide trimeric peptide folds correctly with the correct glycosylation sites that may be needed for any functional activity of the SAg-like motif identified by Bahar group in PNAS.   Therefore, these experiments ( in this manuscript) cannot completely rule out that the SAg-like motif described in SARS -CoV2 does not have any SAg like effects functionally

Finally, while the literature has shown that the putative SAg like motif discovered by Bahar group et al has potential SAg function , indirectly by observing TCR Vbeta skewing in adults and children, it is also very well known that microbial  Superantigen like motifs do not have to have a pure T cell stimulating SAg function but may have several other immune stimulating and autoimmunity inducing activities as well. 

Reviewer 3 Report

In this manuscript, authors try to elucidate whether the SARS-CoV-2 spike protein has superantigenic properties. Based on their results, authors conclude that this is not the case. While I agree that this is the conclusion from the experiments performed, I believe that a simple additional experiment could shed light onto the potential binding capacity of CD28. Please find my recommendation below for additional experiments that I believe will elevate this manuscript to the next level.

Major comments

1.       If authors now show that the Spike protein doesn’t function as a superantigen, but do discuss how it can bind to CD28, what I’m missing are experiments where T cells (primary and/or Jurkat) where stimulated with Spike +/- anti-CD3. In this way, authors could elucidate whether the observed effect in children with MIS-C is due to chronic triggering of co-stimulatory molecules (then probably in combination with a myriad of antigens), which, according to the authors, are upregulated (at least CD28 is according to the results presented here by the authors).

Minor comments

Line 24 – there seems to be a double space after fragment of

Line 26 – there seems to be a double space before Here

Line 72 – verified is a bit odd if there was no activity. Examined would be a better choice.

Line 86 – interesting this ethical approval was obtained in 2019 for studies with COVID. I assume it’s a broad informed consent for use in experiments?

Line 94 and onward – please state dilutions of antibodies used.

Line 110 – If ELISAs were performed according to manufacturer’s instructions, please say so. If any deviations from manufacturer’s protocol were employed, please include.

Line 168 – please capitalize SARS-CoV-2 correctly

Line 172 – remove comma after Fisher R&D

Line 174 – Please change to “After verifying”

Line 178 – remove comma after although

Line 209 – remove “a” before Jurkat